# Automobile Tires' High-Carbon Steel Wire

**Marina Polyakova** [1,*] and **Alexey Stolyarov** [2]

1   Department of Material Processing, Metallurgy, Mechanical Engineering and Materials Processing Institute, Nosov Magnitogorsk State Technical University, 455000 Magnitogorsk, Russia
2   OJSC Magnitogorsk Hardware and Sizing Plant "MMK-METIZ", 455002 Magnitogorsk, Russia; Stolyarov.AY@mmk-metiz.ru
*   Correspondence: m.polyakova@magtu.ru; Tel.: +7-3519-29-84-81

**Definition:** It is a well-known fact that to manufacture an automobile tire more than 200 different materials are used, including high-carbon steel wire. In order to withstand the affecting forces, the tire tread is reinforced with steel wire or other products such as ropes or strands. These ropes are called steel cord. Steel cord can be of different constructions. To ensure a good adhesive bond between the rubber of the tire and the steel cord, the cord is either brass-plated or bronzed. The reason brass or bronze is used is because copper, which is a part of these alloys, makes a high-strength chemical composition with sulfur in rubber. For steel cord, the high carbon steel is usually used at 0.70–0.95% C. This amount of carbon ensures the high strength of the steel cord. This kind of high-quality, unalloyed steel has a pearlitic structure which is designed for multi-pass drawing. To ensure the specified technical characteristics, modern metal reinforcing materials for automobile tires, metal cord and bead wire, must withstand, first of all, a high breaking load with a minimum running meter weight. At present, reinforcing materials of the strength range 2800–3200 MPa are increasingly used, the manufacture of which requires high-strength wire. The production of such wire requires the use of a workpiece with high carbon content, changing the drawing regimes, patenting, and other operations. At the same time, it is necessary to achieve a reduction in the cost of wire manufacturing. In this context, the development and implementation of competitive processes for the manufacture of high-quality, high-strength wire as a reinforcing material for automobile tires is an urgent task.

**Keywords:** high carbon steel wire; reinforcing material; automobile tire; steel cord; bead wire; drawing; patenting; brass-plated wire; laying

## 1. Introduction

Over the past, relatively short, time, the range of reinforcing materials for car tires has undergone significant change. Firstly, this can be explained by the increased requirements for automobile tires, which are now more stringent for mileage, weight, imbalance (power non-uniformity), and so on (Figure 1) [1]. To ensure the elevated technical characteristics of tires, the modern metal cord and bead wire have to withstand a high breaking load with minimum mass of a running meter (linear density), have a sufficient level of bond strength with rubber, and have an increased resistance to fatigue failure under applied loads. To date, special attention is paid to such indicators as the level of residual torsion, straightness, and deflection arrow, which directly affect the manufacturability of the rubber cord sheets' (bead rings) technological processing on modern rubber lines.

The idea of increasing the strength of reinforcing materials for automobile tires and decreasing the volume weight was justified in 1979 through the experience of such leading manufacturers of metal cord and bead wire as Bekaert (Belgium), Goodyear, Firestone (USA), Michelin (France), Bridgestone (Japan), and Pirelli» (Italy) [2]. But the more active process of the substitution of normal-strength reinforced materials for high-strength materials started at the beginning of the 1990s.

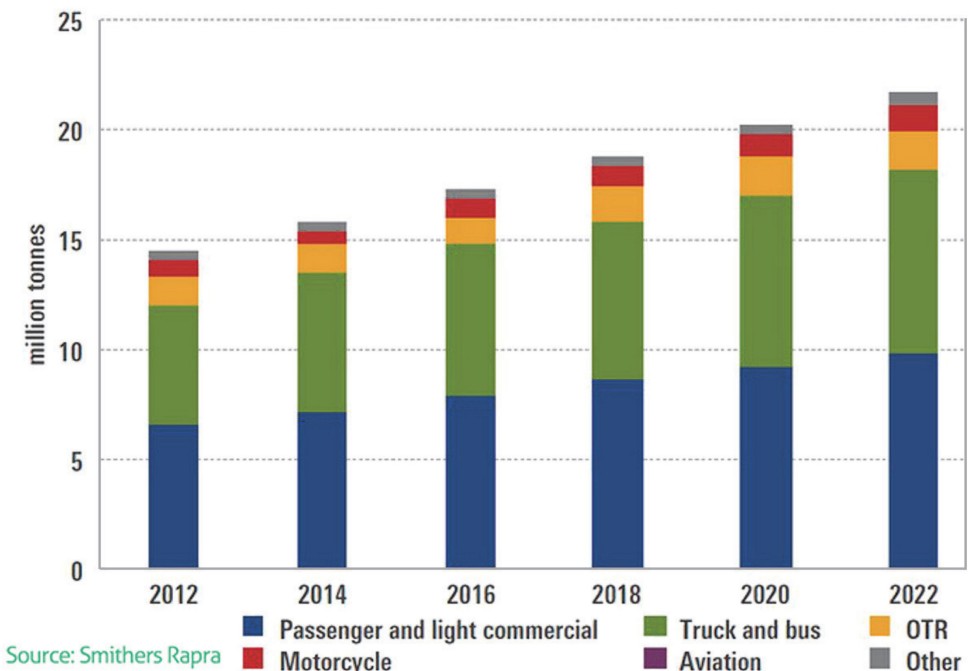

**Figure 1.** Manufacturing of tires for different applications. Reprinted from ref. [1].

While designing new constructions of high-strength steel cord and wires for reinforcing the bead rings of tires, different companies have developed technologies for production wire for high-strength reinforcing materials. The most progressive technologies of manufacturing bead wire were produced in Japan.

At the moment, reinforcing materials of high strength are increasingly used instead of materials with normal tensile (NT) 2400–2800 MPa. They are divided into the following groups: high tensile materials (HT) 2800–3200 MPa and super tensile (ST) materials 3200–3500 MPa. Furthermore, the increase of tensile strength promotes an increased endurance strength of brass-plated wire for metal cord, especially compact beam structures for reinforcing the car tire carcass (Figure 2) [3].

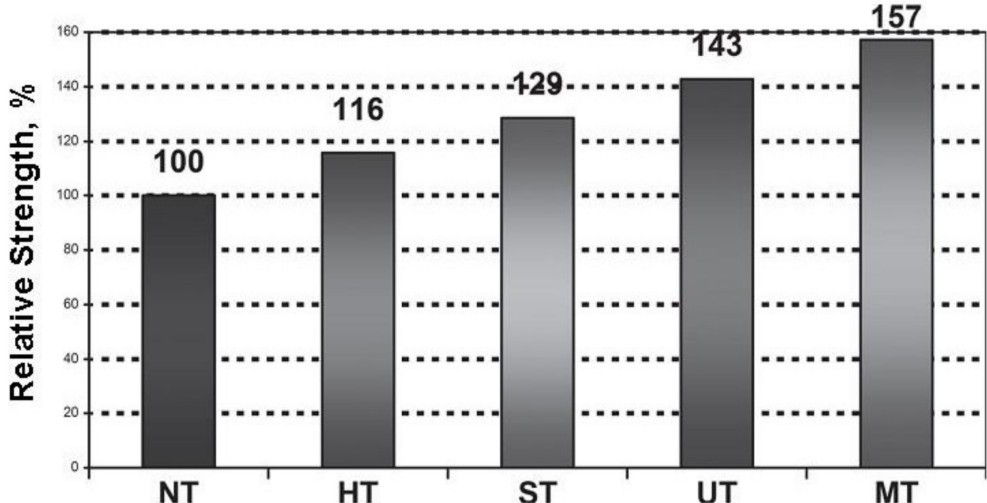

**Figure 2.** Change of wire strength depending on its group. Reprinted from ref. [3].

Figure 3 shows the level of tensile strength of materials which are used for metal cord manufacturing [4].

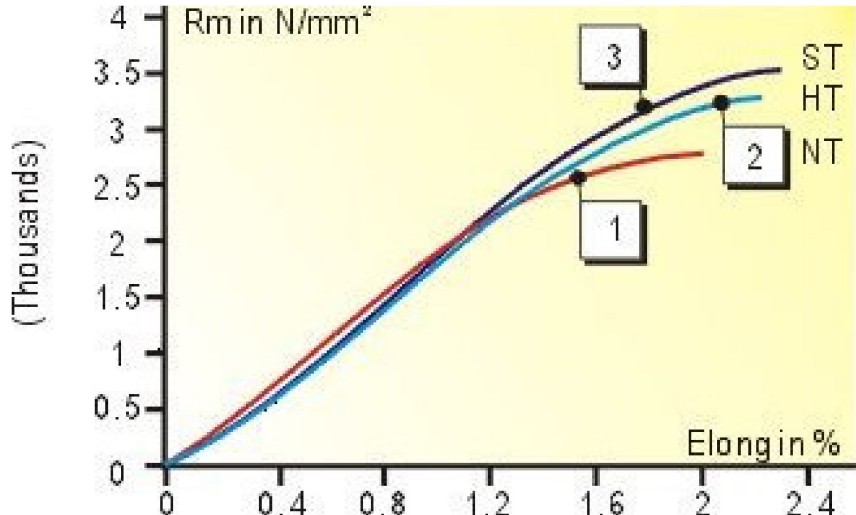

**Figure 3.** Tensile test data for 0.175 mm diameter filaments for (1) normal products, (2) high tensile grade, and (3) an experimental super-high tensile grade. Reprinted from ref. [4].

At present time the tendency to increase the tensile strength of steel cord is observed as shown in Figure 4 [5–8].

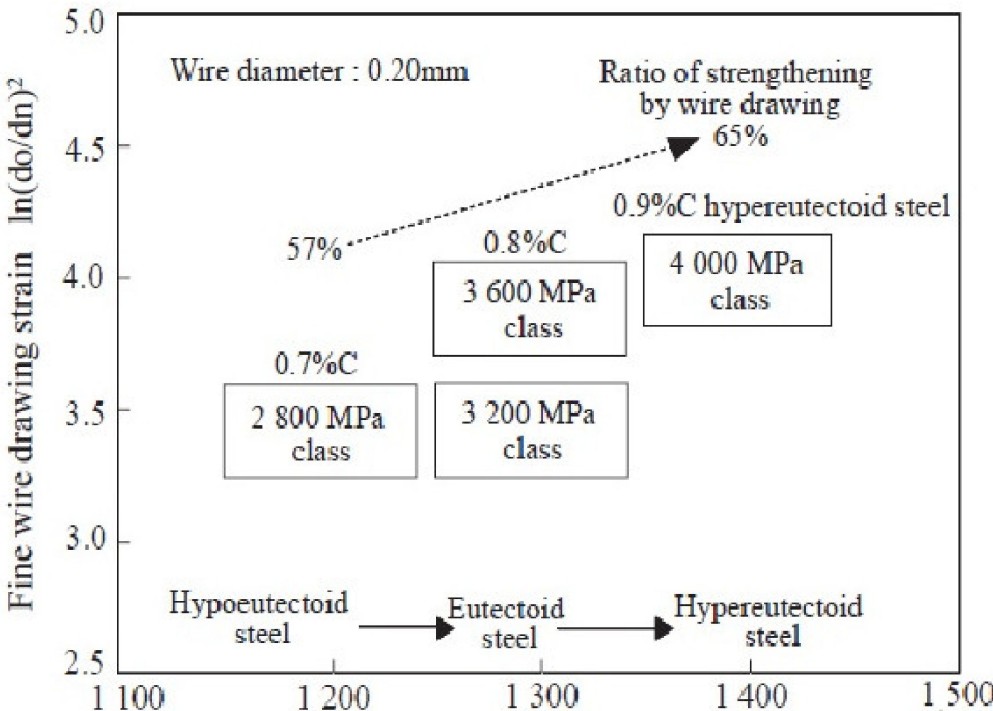

**Figure 4.** Trend of high tensile strength of steel cord. Reprinted from ref. [7].

The tensile strength of metal cord with a diameter of 0.20 mm was 2800 MPa in the 1970s, 3300 MPa in the 1980s, and reached a high strength of 3600 MPa in the early 1990s. The increase of speed and the increase in highway transport demanded a raise in the level of tensile strength to higher values [6,7].

In addition to high tensile strength, the wire for modern reinforcing materials must have a high range of ductile (fatigue) properties. As for metal cord, this condition is necessary to ensure the processability of the double twisting method of high-speed laying. At the level of the HT group (approximately 3000–3200 MPa) thin brass-plated wire with 0.2–0.35 mm in diameter must withstand a certain amount of forward and backward twists.

Otherwise, the quality of the finished steel cord (fatigue endurance) and the productivity of the lay process are sharply reduced.

The aim of this paper is to describe the peculiarities of the manufacturing process of the high-carbon steel wire which is used as the reinforcing material for automobile tires. A general description of the technological process is given in Section 2, which also contains information about special aspects of every technological operation of the manufacturing process. This overview can help the reader to learn about those technological techniques which are necessary in order to produce high-carbon steel wire with the desired exploitation properties. The main tendencies for the improvement of high-carbon steel manufacturing process are denoted in the conclusion.

## 2. Structure, Role, and Demands of the Technological Process of High-Carbon Steel Wire Manufacturing

The main direction of the perspective technological design and development of new technological processes in metallurgy is the creation of such technological systems which are based on low-operational, unmanned, and waste-free technology providing a multiple increase in labor productivity and a significant improvement in product quality and other indicators.

The technology for the manufacture of high-strength wire for automobile tires should be generally observed. For example, in Japan, there are conceptually two main directions to achieve the required level of steel-wire strength: strengthening in patenting and strengthening in drawing. Moreover, these two directions are each esteemed comprehensively in terms of the regularity of the pearlite structure refinement in the wire [9–11].

At the input stage of the technological process of manufacturing wire for reinforcing materials for automobile tires, there are main and auxiliary materials (high-carbon steel wire rod, copper and zinc anodes, etc.), and at the output of the process there is the cold-deformed (brass plated, bronzed) wire.

The structure of the actual technological process of manufacturing wire for the reinforcement of materials for automobile tires consists of the following main interrelated subprocesses:

- surface preparation and "rough-medium" drawing of sorbitized wire rod;
- patenting of cold deformed workpiece. Patenting can be a transitory operation used for the formation of the final properties of the wire;
- brass plating to ensure the adhesion of the metal cord to rubber;
- finishing drawing;
- laying of the metal cord;
- annealing, processing in alternative deformation (if necessary for bead rings of tires).

The technological scheme can be presented by means of blocks. Each block contains information about the name of the technological operation. The technological scheme for brass-plated, high-carbon steel wire with high strength for steel cord is presented in Figure 5, as are the range of diameters of the processed wire. For thin high-strength brass plated wire with 0.85% C the diameters of patented workpiece were chosen as shown in the blocks. Based on the experimental results, it was proved [12] that the intermediate operation of patenting was obligatory in the manufacturing process because it reduced wire breakage in drawing.

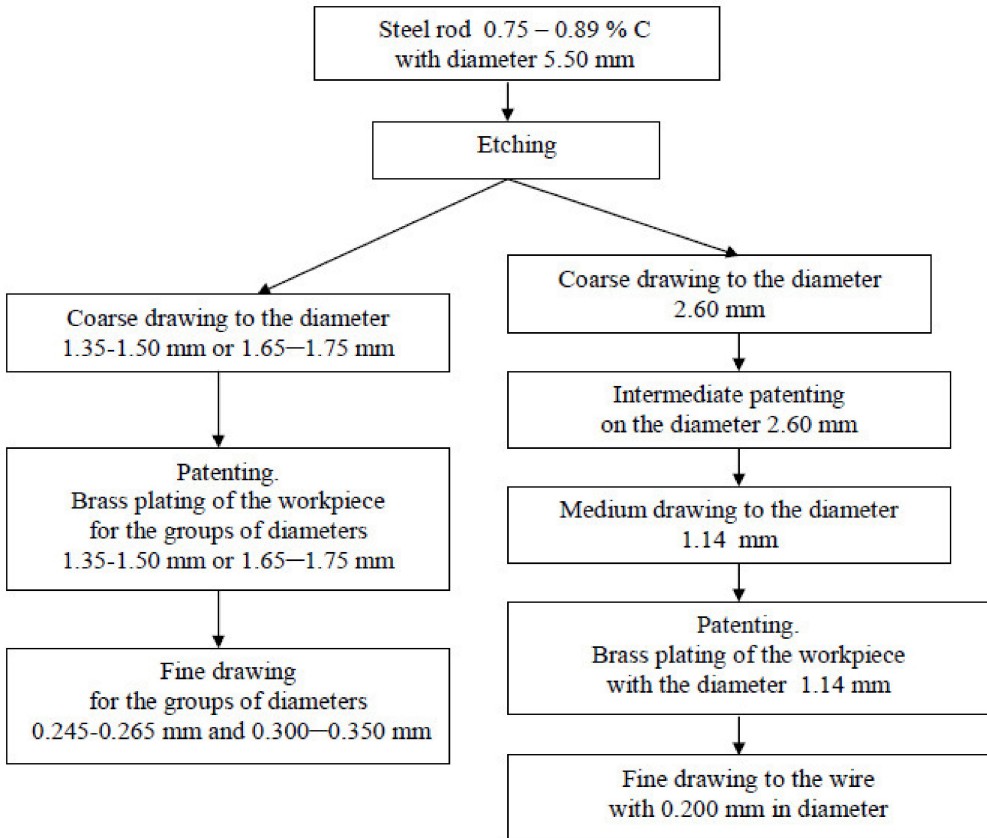

**Figure 5.** Technological scheme for the manufacture of brass-plated, high-carbon steel wire with high strength for steel cord.

At the present time high-strength, bronzed-steel wire with a diameter of 1.60 mm is highly requested for all-steel automobile tires. The current way to get the desired level of mechanical properties is to use a special kind of heat treatment in an air-fluidized bed with alternative bending as final operations. The application of these kinds of processes guarantees a ratio of yield strength to tensile strength of 75–85%. The technological scheme for bronzed, high-carbon steel wire is presented in Figure 6.

The implementation of these technological schemes (see Figures 5 and 6) at the industrial scale makes it possible to improve the competitiveness of the manufactured high-strength steel wire for cord [12].

### 2.1. Steel Rod for High-Strength Wire Manufacturing

The choice of steel rod for the manufacture of high-strength wire has a significant role in the technological process of the production of reinforcing materials for automobile tires [13]. One of the basic factors which affects the technological effectiveness of metal cord manufacturing, as well as its technical and exploitation characteristics, is the quality of the high carbon steel rod which is used as a reinforcing material in automobile tires. Demands on the steel rod for metal cord and bead wire are formulated, first of all, taking into consideration further regimes of its processing and the functions of the final product.

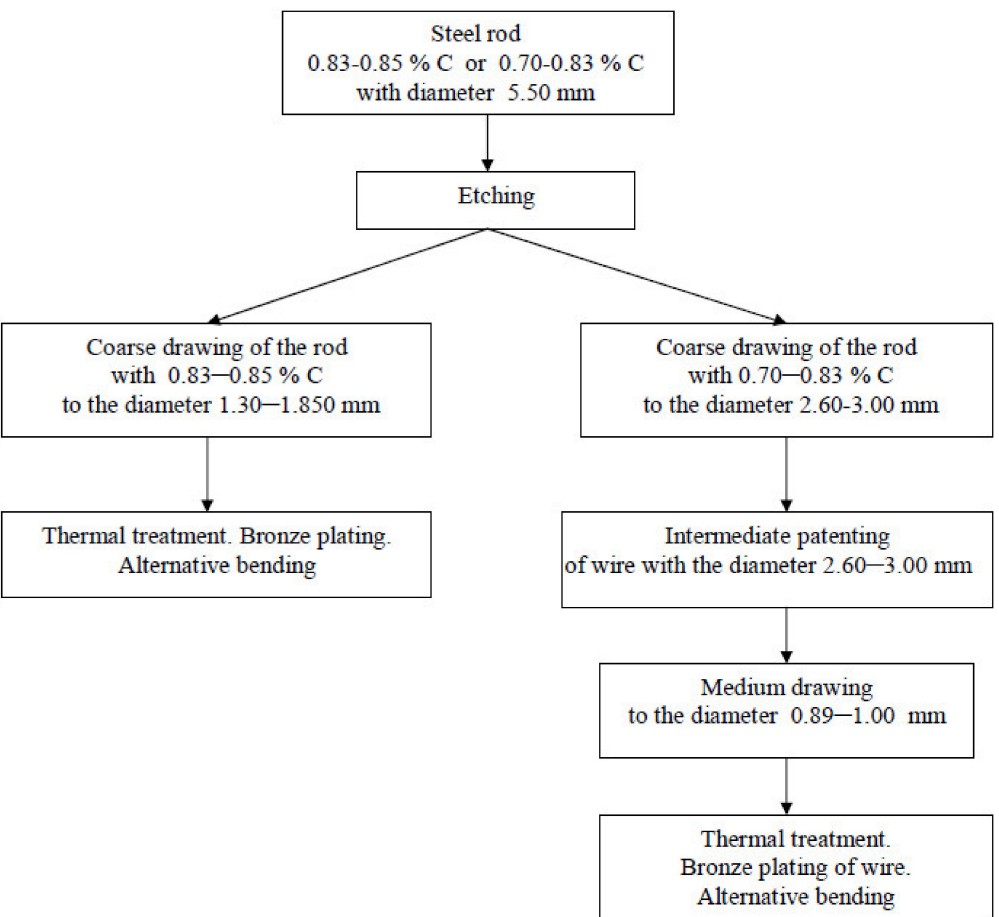

**Figure 6.** Technological scheme for the manufacture of bronzed, high-carbon steel wire with high strength.

To manufacture high-strength and ultra-high-strength metal cord, steel rod made from high-carbon steel with 0.70–0.95% C and 5.5 mm in diameter is used. The pearlitic microstructure is typical for steel with such an amount of carbon and consists of ferrite-carbide mixture (Figure 7).

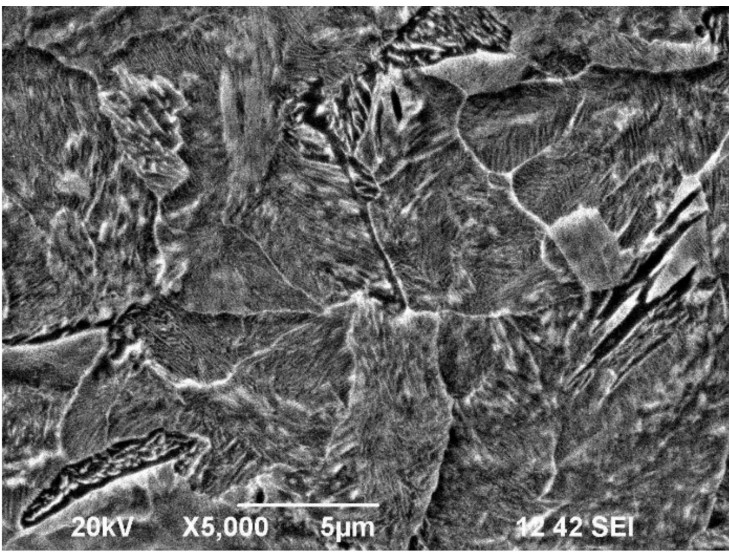

**Figure 7.** Microstructure of steel rod 0.70% C for metal cord manufacturing before drawing.

As shown in Figure 7, the microstructure consists of troostite with small amount of bainite, and ferrite which is located as a net around pearlite colonies

Special demands are exhibited to the chemical composition of steel, the quantity of impurities and imperfections in steel, and the macro and microstructure. To ensure the required level of properties such companies as Cobe Steel [14], Nippon Steel [15], Kawasaki Steel (Japan) [16], THYSSEN (Germany) [17], and others alloy their steel with chromium, copper, manganese, cobalt, etc.

It is stated in many papers [18–24] that chemical composition, pollution of steel by non-metallic inclusions, results of liquation processes, presence of scale on the surface of rod and its decarburization, and the peculiarities of macro and microstructure have a great influence on the processability of steel rod in the following operations of technology: rough drawing, patenting, drawing of brass-plated wire, and laying, as well as the quality of the final product.

### 2.2. Role of Drawing in the Technological Process of High-Strength Wire Manufacturing

In drawing, the cross section of a long rod or wire is reduced when it is pulled through a die. Tensile strain and compression strain are obvious in drawing. The major processing variables in drawing are reduction in cross-sectional area, die angle, friction along the die-workpiece interface, and drawing speed. Drawing is usually performed as a cold working operation. Drawing speeds are as high as 50 m/s for steel cord. In drawing, reductions in the cross-sectional area per pass range up to about 45%. Usually, the smaller the initial cross section, the smaller the reduction per pass. Fine wires for steel cord usually are drawn at 15 to 25% reduction per pass. In order to avoid the breakage of wire in high-speed drawing, the emulsion coolant "oil in water" is used.

In metal cord manufacturing it is impossible to produce high carbon steel wire with a diameter less than 1 mm directly from the rod because of the large amount of total reduction in drawing [25]. For this reason, the technological process «Rod—Wire for Metal Cord» is divided into several subprocesses and can be presented as the combination of basic operations of drawing in monolithic dies and thermal treatment (patenting).

Conditionally it can be determined as two variants:

1. rod—workpiece for the final wire (rough process stage)
2. brass-plated wire after patenting—thin bras-plated wire (final process stage).

As a matter of fact, the rough process stage «Rod—Workpiece for the Final Wire» is the shape-generating stage which ensures the necessary diameter of the workpiece for the further drawing of the rod so as to manufacture the final wire with the definite diameter. To lower costs for the rough process stage, it is necessary, on the one hand, to reduce the quantity of thermal treatments and, on the other, it is necessary to keep in mind that with the increase of the total deformation degree the probability of breakage of the wire in drawing also raises. In particular, cracks, tears, and other kinds of breakage are dangerous because these kinds of defects do not disappear during further heat treatment and decrease the wire quality as well as the metal cord laid from this wire.

In the manufacture of bead wire with a diameter of between 1.30–1.85 mm at present time both physical and chemical properties of the final product are dependent on the process stage «Rod—Workpiece for the Final Wire». For this reason, special attention is paid to the regimes of coarse drawing in the technological process of bead bronzed wire.

The role of the final process stage (fine drawing) in the manufacture of metal cord, besides shaping, is of ensuring the strength and ductile properties of the final wire. This is why the diameter of the workpiece for the final wire is chosen by taking into consideration the necessary degree of total deformation. The key points in this case are the steel composition (carbon content), the degree of total deformation (determination of the diameter of patented brass-plated workpiece), and the regimes of wet drawing. In drawing, pearlite colonies of the processed high-carbon steel wire elongate towards the drawing direction as shown in Figure 8. This kind of microstructure is characterized by a disposition of grains along the force applied.

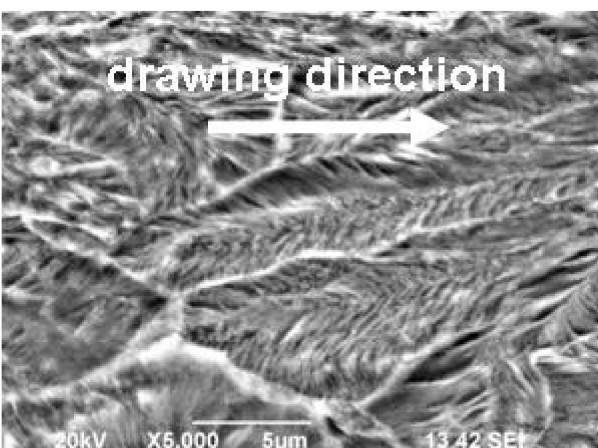

**Figure 8.** Microstructure of high-carbon steel wire 0.70% C after drawing.

As compared with coarse drawing, the fine drawing of brass-plated wire is characterized by tough friction conditions and a higher drawing rate. The approaches used for the designs associated with drawing of high-strength wire for metal cord are presented in [26–28]. Quality control in drawing is based on the distribution of hardness across the wire including fine brass-plated wire. The difference in hardness between outer and internal areas of wire should not be more than 7% [26]. This is why, besides the magnitude of reduction, the control factor to ensure the properties of cold-drawn wire is the angle of the drawing tool.

In the drawing of high-carbon steel wire, much attention is paid to the negative affect of the heat deformation warming-up. It is considered that the temperature of the wire on the finishing drum of the drawing mill should not be more than 150 °C. The negative affect of the temperature is proved during tribological analysis of the contact system «Brass-Lubricant-Drawing Tool» which was carried out by specialists of «Michelin» (France) [29].

For the coarse drawing of wire, the direct-flow drawing mills with intensive system for cooling drums and drawing tools are used. Well-known drawing mills are produced by «GCR EURODRAW SPA» (Italy), «MARIO FRIGERIO SPA COMPANY» (Italy), «ERNST KOCH GMBH @ CO.» (Germany), «SWARAJ TECHNOCRAD PVT. LTD.» (India). Special attention is paid to the quality of the surface of motoblocs.

For wet drawing of brass-plated wire, the drawing mills of higher deformation ratio produced by «M + E Macchine + Engineering S.p.a», «VVM», «Team Meccanica», and «Samp Steel» (Italy) are used. Drawing mills are equipped with a high pressure emulsion supply system [30] and cooling for the drawing tools, fine dies, and drawing drums [31]. The maximum drawing rate reaches 20–25 m/s. The drawing emulsion is fed to the group of mills through a closed loop, which make it possible to effectively control its parameters.

It is known [32] that an increase in drawing rate leads to a reduction of the viscosity of the lubricant and the thickness of its layer in the deformation zone. As a result, the wear of the drawing tool increases and the warming-up of the wire and pulling pulleys of the drawing mill intensify. This should be taken into consideration when designing the regimes of wet drawing for thin high-carbon steel wire on sliding drawing mills.

The increase of drawing rate also facilitates the localization of deformations on the outer layers of the wire and, eventually, an irregularity across the wire cross section. This fact enhances the influence of surface phenomena during the wet drawing of brass-plated wire, in other words, it enhances the influence of the scale factor.

It has been stated [25,33–35] that when drawing high-carbon steel wire, the development of dynamic and static deformation aging processes leads to a deterioration in the plastic and fatigue life of thin brass-plated wire. For this reason, in drawing thin brass-plated wire on drawing mills of wet drawing, lower deformation degrees are used as compared with coarse drawing. Wire slip on the pull pulleys of the drawing mill is the

result of additional thermal effects on the wire. Taking into consideration the negative effect of temperature in drawing thin brass-plated wire it is necessary to reduce wire slip on the final passes as well as single reductions [36] and ensure the effective cooling of the wire in its exit from the finishing die.

### 2.3. Role of Thermal Treatment in the Technological Process of High-Strength Wire Manufacturing

There are two kinds of thermal treatment in the technological processes of metal cord and bead wire manufacturing. Patenting is used to recover ductility of cold-drawn wire and to ensure the necessary level of mechanical properties in the final product. Annealing of the final bead wire is used for stress relaxation which is necessary to match the requirements of normative and technical documentation to the relative elongation values of the finished wire. In both cases, a reliable and efficient implementation of the temperature regime is required, which provides not only the required complex of properties for the finished product, but also a minimal energy consumption for the operation.

Analysis of the applied technologies of patenting shows that to get the desired microstructure, air cooling, heating (cooling) in fluidized area of particles, quenching in water, keeping temperature by the direct transmission of electric power, etc. are used [26,37–42].

There are two variants of practice in patenting. In the first case, the cooling rate is regulated only by the difference of temperatures between the heating of the wire in a furnace and a bath of isothermal decomposition; while other parameters also affect the cooling rate, in particular the coefficient of forced convective heat transfer between the wire and the bath environment, they are not taken into account. The other way is to consider both the difference of temperatures between the heating of the wire in a furnace and a bath of isothermal decomposition and the coefficient of forced convective heat transfer between the wire and the bath environment. Special attention to this aspect is paid in [43–45] where different methods which ensure the reliable regime of wire cooling are described.

With regard to the process of patenting wire in lead, the efficiency of convective heat transfer during the decomposition of supercooled austenite can be increased by raising the speed of movement of the lead (wire).

More perspectives, from the point of view of energy saving, ecology, and harmful effects on the human body, for methods of wire heating and cooling can be used not only in patenting but also during annealing of finished bead wire. In particular, fluidized bed heating technology, which, with proper technical support, has a number of advantages over heating in lead is widely used to heat the wire to 450–500 °C.

### 2.4. Deposition of Adhesive Coatings on the Metal Cord Wire

To date, the assortment of wire for tire bead ring reinforcement has become wider with a consequent substitution of brass-plated wire for bronze-plated wire which is considered to be more competitive [46,47]. Technological schemes of bead brass-plated wire and bronze-plated wire are different. Brass-plated coating is deposited on the wire by the consistent electrochemical deposition of the copper layer and the zinc layer. In this case, it is necessary to heat the wire to initiate the diffusion process of copper and zinc. The technological process of bronze deposition is more efficient. Bronze is deposited chemically by means of simultaneous deposition of copper and tin in one bath. One of the disadvantages of bronze coating, as compared with brass-plated coating, is its low level of adhesion with rubber. However, the technology of preparation of rubber mixtures at tire manufacturing enterprises makes it possible to change this parameter through a correction of the compounding. As a result, adhesion of the bronzed wire increases which allows it to be used quite successfully for reinforcing the bead rings of tires.

Taking into consideration manufacturing costs together with the level of exploitation properties the perspective way is to carry out the industrial technology of deposition of bronze coating on bead wire instead of brass-plating.

*2.5. Use of Setups for Alternative Bending to Increase the Ductility of Bead Wire*

The application of enterprises for tire manufacturing using modern high-capacity bead-making units has led to the formulation of strict demands to the bead wire mechanical properties. As a result, the percentage ratio of yield strength to tensile strength should be equal to 75–85% in accordance with the demands in technical certificates to the bead wire. This can be ensured by alternative bending of cold drawn wire.

Alternative bending causes the appearance of stresses which lead to the breakup the unstable substructures in the processed wire [48,49]. This kind of processing promotes the increase of its ductile properties.

*2.6. Laying*

The laying of metal cord is basically carried out on single twisting machines when the wire is not exposed to alternating deformation. For this reason, the existing reserve of plasticity in the wire ensures a sufficient level of its manufacturability in laying. Breakage of the wire in laying can be predominantly explained by the presence of non-metallic inclusions in the steel [50].

Machines operating on the principle of double twisting, when metal cord is twisted in two pitches during one rotation of the rotor, are usually used for laying. Laying on double twisting machines is more efficient and effective as compared with the same operation when single twisting rotor type machines are used. But at the same time, thin brass-plated wire is exposed to high alternative deformation, hence it raises demands to the mechanical properties [51].

**3. Conclusions and Prospects**

The manufacturing process consists of complex technological actions on the workpiece. During any technological operation the workpiece changes its parameters. Furthermore, products made of modern materials can be processed technologically in a number of different ways. Under such conditions the manufacturer should have some algorithms and models to select the technological process considered to be optimal taking into consideration the peculiarities of the industrial enterprise. This technological process has to guarantee the production of the finished product with the related level of quality and exploitation properties.

Because of high strength and high corrosion resistance, the steel cord still remains the main reinforcing material for tires of different types of automobiles. New trends in steel cord manufacturing processes are presented in [3,52,53]. The necessity to decrease artificial pollutants in the use the gasoline engines put forward new tasks for engineers to find new ways to increase the steel cord tensile strength. One of the prospective ways is to use steel with a nanostructure which ensures high values of both tensile strength and ductility in the processed material [54–57]. At the present time, the implementation of methods of severe plastic deformation under industrial conditions is on the cutting edge of technological progress. However, considering that the diameter of steel wire for cord is less than 1 mm, it would be necessary to create alternative ways to achieve a similar nanostructure in the processed material.

The technological process of steel cord manufacturing consists of several operations of different physical natures. For this reason, the risk of breakage of the processed material increases. This is why one of the important problems of the manufacturing process is to decrease the quantity of non-metallic inclusions, segregation of alloyed elements in steel for cord, surface blemishes, etc. Engineering should be addressed to solve these issues.

Perspectives for the design of the manufacturing process for a competitive high-carbon steel wire for steel cord and bead wire should be based on the solution of the following tasks:

- development of a methodology for calculating the drawing modes of high-carbon wire based on the selection and use of the fracture criterion and assessment of the influence of the deformation zone shape factor on its destruction;

- investigation of the nature of metal flow in the near-surface layer during drawing, assessment of the influence of drawing factors on its depth, and development of practical recommendations for calculating deformation modes of thin high strength brass plated, and bronze coated wire for metal cord and bead wire.

Furthermore, the level of technology of every manufacturing process has a decisive influence on its economic performance. This is why the choice of the optimal variant of the technological process should be carried out on the basis of the most important indicators of its effectiveness: productivity, cost, and quality of products. The tendency to find new materials to substitute steel wire for automobile tires with the required level of exploitation properties remains a challenge for scientists and engineers.

**Author Contributions:** Conceptualization, M.P.; project administration, A.S. All authors have read and agreed to the published version of the manuscript.

**Funding:** This research received no external funding.

**Conflicts of Interest:** The authors declare no conflict of interest.

**Entry Link on the Encyclopedia Platform: :** https://encyclopedia.pub/14524.

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
