# Peer review of "Automobile Tires’ High-Carbon Steel Wire"

_encyclopedia, doi:10.3390/encyclopedia1030066_

Round 1
Reviewer 1 Report
The article is written in a very "professor´s" style. A greater depth of information is needed in the text, which would at least support:
- schematic images of the microstructure (possibly modified by original photographs)
- tensile diagrams revealing the nature of the difference between normal tensile - high tensile - super tensile materials.
- Furthermore, a brief explanation of tribological processes in wire formation,
- Information on coolants when drawing wires, ...
Deeper documentation of the physical nature of deformation processes while achieving high strength values ​​would then also help the formulation of conclusions (now they are generally formulated about unclear market requirements, necessary flexibility of manufacturers, ...): what is still area to increase the strength properties of reinforcing wires, what research and development can be expected in the future
Fig.3 and 4 must be commented in more detail (what does each branch of the block diagram represent?) And placed already in chap. 2.
Terminology:
The authors state the need to maintain ductility at high strength of reinforcing wires and fatigue strength - more precisely, it is a cyclic plasticity

Author Response
Authors thanks to the Reviewer for the corrections. All of them were taken into consideration to improve the paper

Reviewer 2 Report
The authors discussed the production of high-strength wire that requires high carbon content, special drawing procedure, patenting, and other unique operations. The work is very important, but authors may pay an attention to the format of the manuscript. For instance, the authors used different line spacing in different part of the manuscript without explanation of the reasons. The authors may also make the review broader.
p4, line 101-105: Authors stated, “To ensure the required level of properties such companies as «Cobe Steel», «Nippon Steel», «Kawasaki Steel» (Japan), «THYSSEN» (Germany) and others use alloying of steel by chromium, copper, manganese, cobalt [9].”
Citation 9 is “Fetisov, V.P. Deformational hardening of carbon steel; Publisher: Mir, Moscow, Russia, 2005; 197 p. “It is unclear from this paragraph who invented the technology for addition of chromium, copper, manganese, cobalt, but it sounds that Fetisov invented all these. Authors should cite the work from those companies if the invention was done in those companies. In an addition, other alloying element additions to high-carbon rod should be mentioned. Typical citations are:
Modification of Type B Inclusions by Calcium Treatment in High-Carbon Hard-Wire Steel, By Wang, LZ (Wang, Linzhu) 1, 2, Xi, ZB (Xi, Zuobing) 1, 2, Li, CR (Li, Changrong) 1, 2, METALS, Volume11, Issue5, Article Number676, DOI10.3390/met11050676, PublishedMAY 2021
Mechanical Performance-Based Optimum Design of High Carbon Pearlitic Steel by Particle Swarm Optimization By Qiao, L (Qiao, Ling), Wang, ZB (Wang, Zibo), Wang, Y (Wang, Yuan) 2, Zhu, JC (Zhu, Jingchuan) 1, STEEL RESEARCH INTERNATIONAL, Volume92, Issue1, Article Number2000252, DOI10.1002/srin.202000252, PublishedJAN 2021
Torsion performance of pearlitic steel wires: Effects of morphology and crystallinity of cementite, By Zhou, LC (Zhou, Lichu) 1, Fang, F (Fang, Feng) 1, Wang, LP (Wang, Liping) 1, Hu, XJ (Hu, Xianjun) , Xie, ZH (Xie, Zonghan) ,Jiang, JQ (Jiang, Jianqing) 1, MATERIALS SCIENCE AND ENGINEERING A-STRUCTURAL MATERIALS PROPERTIES MICROSTRUCTURE AND PROCESSING, Volume743, Page425-435, DOI10.1016/j.msea.2018.11.113
Effect of Nb micro-alloying on microstructure and properties of thermo-mechanically processed high carbon pearlitic steel, By Dey, I (Dey, I.) 1, Chandra, S (Chandra, S.) 1, Saha, R (Saha, R.) 2, Ghosh, SK (Ghosh, S. K.) 1, MATERIALS CHARACTERIZATION, Volume140, Page45-54, DOI10.1016/j.matchar.2018.03.038. PublishedJUN 2018
Optimization of mechanical properties of high-carbon pearlitic steels with Si and V additions, By Han, K (Han, K) , Edmonds, DV (Edmonds, DV) , Smith, GDW (Smith, GDW), METALLURGICAL AND MATERIALS TRANSACTIONS A-PHYSICAL METALLURGY AND MATERIALS SCIENCE, Volume 32, Issue 6, Page 1313-1324, DOI10.1007/s11661-001-0222-7, Published JUN 2001
DEVELOPMENTS IN ULTRA-HIGH-CARBON STEELS FOR WIRE ROD PRODUCTION ACHIEVED THROUGH MICROALLOYING ADDITIONS, By HAN, K (HAN, K) , SMITH, GDW (SMITH, GDW) , EDMONDS, DV (EDMONDS, DV), MATERIALS & DESIGN, Volume14, Issue1, Page79-82, DOI10.1016/0261-3069(93)90054-Y, Published FEB 1993
The authors can pay an attention to some grammar issues; For example, in line 126 “there are no any specific demands to the strength properties of the wire …..”. I also don’t agree with the science of last sentence and the following sentence in line 126-127 “That is why the operation of patenting serves only for the recovery of ductile properties of the wire and can be resented as so called technological barrier. ” For example, it is undesirable if the patent process leads to the spheroidization in cementite.
Page 7, line 243-247: This paragraph is difficult to read.
Figures 3 and 4: It is unclear where is new in the steps in these two figures. The sizes and chemistries are very specific. Have they been optimized?
Some citations are incomplete and difficult to find.
Author Response
Authors thanks to the Reviewer for corrections. All of them were taken into consideration to improve the paper.

Reviewer 3 Report
- The topic addressed in the paper is of current interest and in line with the profile of the journal Encyclopedia and the disciplines likely to be addressed in the papers proposed for publication.
- The authors of the paper have the necessary experience to elaborate a paper corresponding to the objectives (aims) of the journal. They have previously published results of research conducted in areas close to that addressed in the paper.
- As the title of the paper is “Wire carbon steel wire”, I think that both in the Abstract and in the paper other areas in which carbon steel wire could be used could be indicated (in the Abstract, only the use for car tires is mentioned). The title change (High carbon steel wire for car tires?) could also be considered.
- It is preferable to use ISO-promoted units of measurement (MPa instead of N/mm2).
- Although the title of the paper also mentions "high carbon steel", there is no mention of the carbon contents of steels included in this category.
- The fact that “Patenting is used to recover ductility” is mentioned twice in the text of the article.
- The first statement in the chapter on conclusions is not generally valid {“The manufacturing process is the basics of the activity of every enterprise.”). There are enterprises where the core basics are not manufacturing processes.
- The first statements in Chapter 3 are of a general nature and do not take into account the subject of the paper. The statement also applies to the last paragraphs of Chapter 3; the content of these paragraphs is little connected with the subject of the paper.
- The graphical representations in Figures 3 and 4 could be commented on in a separate chapter and not in the conclusions chapter.
- At the end of the Introduction, some information regarding the structure of the paper could be included, with a highlighting/classification of the issues that will be addressed in the paper.
- An introduction of graphical representations highlighting the various aspects of the use/manufacture of high carbon steel wire would be useful.
- Over 20 of the 50 works included in the reference list seem to have authors from Russia or the former Soviet Union. A Google Scholar search using the concept of “High Carbon Steel Wire” highlights the fact that there are still many works, with many citations and that were not used by the authors of the paper:
For example:
- a) Tarui, T .; Maruyama, N. Microstructure control and strengthening of high-carbon steel wires. Nippon Steel Technical Report No. 91 January 2005, https://www.nipponsteel.com/en/tech/report/nsc/pdf/n9112.pdf (cited by 118);
- b) Kazeminezhad, M .; Karimi Taheri, A. The effect of controlled cooling after hot rolling on the mechanical properties of a commercial high carbon steel wire rod. Materials & Design
2003, 24 (6), 415-421 (Cited by 61)
- c) Kemp, I. P .; Pollard, G .; Bramley, A. N. Static strain aging in high carbon steel wire.
Materials Science and Technology 1990, 6 (4) (cited by 35), etc.
- Authors need to pay more attention to writing the reference list. In some cases, it is not clear whether it is a paper or a book (see references to numbers 13, 14, 44). Journal names should not be capitalized. Italics should be used for all journals titles (see journals titles from references 29, 38, 47, etc.). Book titles should also be written in italics. In the case of patents, the names of the authors are not specified, but there is much-unsolicited information in a list of bibliographic references. In the case of reference no. 1, the title of the site is missing. In the case of books, the numbers of those pages to which reference is made are requested and not the total number of pages of the books.
- Authors need to pay more attention to editing the paper. Thus, at the end of the statement “For this reason, special attention is paid to the regimes of coarse drawing in the technological process of bead bronzed wire” a dot is required as a punctuation mark.
In the manufacturing processes, it is customary to use the concept of "workpiece", instead of "semi-product".
It would be useful to correct the text of the paper by a person who knows English better (there are many situations where commas seem to be missing or misused). For example, commas could be placed after the expressions “As a result”, “For this reason”, “But at the same time”, etc.). In the wording “There two kinds of thermal treatment in the technological processes of metal cord and bead wire manufacturing.”, the verb “are” seems to be missing.
The last paragraph of section 2.2 (but also other paragraphs in the paper, for example, the last paragraph of section 2.3) seems to have been edited with a different line spacing.
Author Response

(The authors gave the same response as above.)

Round 2
Reviewer 1 Report
The quality of the article has noticeably increased.
Author Response
Open Review
Comments and Suggestions for Authors
The quality of the article has noticeably increased.
Answer.
Authors are thankful to the Reviewer for high estimation of the paper.
Reviewer 2 Report
I appreciate that the authors made corrections and improve the manuscript. It reads much better. However, I still see rooms for improvement. For example, the authors could improve their grammar further and make the manuscript clearer. Below are some examples.
It appears that Figure 5 has been published before, but the figure is very specific for certain cord, and is not always clear and consistent with the text. The numbers given in the figure need to be explained if they are so specific. For example, why the first patent is done for 3.0 mm wire? If authors want to use phrase “final patenting”, the other one should be “initial patenting” or similar. Why the manufacture use strands rather than wire with larger diameter? Authors need to explain why the information in chart for figure 5 is inconsistent with figure 6, or use only one figure.
Line 209-211: Authors stated “In drawing pearlite colonies of the processed high carbon steel wire elongate towards the drawing direction as it is shown in Figure 9. This kind of microstructure is denoted as texture which is characterized by disposition of grains along the force applied.” In metallurgy, texture usually means the crystallographic texture, different from what authors stated. I therefore don’t agree with authors said here.
Line 312-316: This paragraph has been improved but still has errors and is unclear. “Alternative bending causes the appearance of stresses which lead to the breakup the unstable substructures in the processed wire. These substructures are typical to the cold-hardened high carbon steel [50, 51]. Hence, dislocation kind of substructures transforms to the disclination substructure. It decreases the gradient of internal stresses in the wire and promotes the increase of its ductile properties. “
Author Response
Authors are thankful the Reviewer for remarks and comments. Authors followed the advices of the Reviewer to improve the paper.
Comments and Suggestions for Authors
I appreciate that the authors made corrections and improve the manuscript. It reads much better. However, I still see rooms for improvement. For example, the authors could improve their grammar further and make the manuscript clearer. Below are some examples.
It appears that Figure 5 has been published before, but the figure is very specific for certain cord, and is not always clear and consistent with the text. The numbers given in the figure need to be explained if they are so specific. For example, why the first patent is done for 3.0 mm wire? If authors want to use phrase “final patenting”, the other one should be “initial patenting” or similar. Why the manufacture use strands rather than wire with larger diameter? Authors need to explain why the information in chart for figure 5 is inconsistent with figure 6, or use only one figure.
Answer. Figure 5 shows the technological process of steel cord manufacturing which was published in R&D Kobe Steel Engineering Reports 2000, 50(3), 32-38 by Minamida, T. et al. To Authors’ opinion it would be not correct to change anything in this figure. At the same time it would be difficult to explain any numbers because this technological scheme is developed by specialists from Japan. Authors will follow the advice of the Reviewer and delete this figure from the text. Numbers of all figures which are presented further in the text were corrected also.
Line 209-211: Authors stated “In drawing pearlite colonies of the processed high carbon steel wire elongate towards the drawing direction as it is shown in Figure 9. This kind of microstructure is denoted as texture which is characterized by disposition of grains along the force applied.” In metallurgy, texture usually means the crystallographic texture, different from what authors stated. I therefore don’t agree with authors said here.
Answer. Authors agree with the Reviewer. This sentence was corrected in the text in the following way.
This kind of microstructure is characterized by disposition of grains along the force applied.
Line 312-316: This paragraph has been improved but still has errors and is unclear. “Alternative bending causes the appearance of stresses which lead to the breakup the unstable substructures in the processed wire. These substructures are typical to the cold-hardened high carbon steel [50, 51]. Hence, dislocation kind of substructures transforms to the disclination substructure. It decreases the gradient of internal stresses in the wire and promotes the increase of its ductile properties. “
Answer. The paragraph was paraphrased in the following way:
Alternative bending causes the appearance of stresses which lead to the breakup the unstable substructures in the processed wire [50,51]. This kind of processing promotes the increase of its ductile properties.
Reviewer 3 Report
1. The authors have improved the content of the article.
2. In some places in the paper, free spaces are required between the symbol corresponding to the percentage content (“%”) and the carbon symbol. It is necessary to write “% C” instead of “%C”.
3. Commas are required in the forms "For steel cord, the high carbon steel is usually…", "In order to avoid the breakage of wire in high-speed drawing, the emulsion", etc.
4. It may be written “about those technological techniques which are necessary to use in…”, instead of “about those technological techniques which is necessary to use in…”, “of bainite and ferrite”, instead of “of beinite and ferrite”.
Author Response
Comments and Suggestions for Authors
- The authors have improved the content of the article.
- In some places in the paper, free spaces are required between the symbol corresponding to the percentage content (“%”) and the carbon symbol. It is necessary to write “% C” instead of “%C”.
- Commas are required in the forms "For steel cord, the high carbon steel is usually…", "In order to avoid the breakage of wire in high-speed drawing, the emulsion", etc.
- It may be written “about those technological techniques which are necessary to use in…”, instead of “about those technological techniques which is necessary to use in…”, “of bainite and ferrite”, instead of “of beinite and ferrite”.
Response: Authors are thankful the Reviewer for careful attention to the paper. The paper was corrected taking into consideration the remarks.
Round 3
Reviewer 2 Report
I appreciate that authors made improvement of the manuscript. Although I think that the authors can improve the manuscript further, different people may have different opinion. I therefore will leave the authors to make decision if they want to improve further in their final form.
Reviewer 3 Report
I agree with the publication of the paper.